# Comparison of Postoperative Ovarian Reserve Function Following Laparoscopic Hysterectomy and Laparoscopic Myomectomy: A Prospective Comparative Pilot Study

**DOI:** 10.3390/jcm10143077

**Published:** 2021-07-12

**Authors:** Hye-Yon Cho, Min Sun Kyung

**Affiliations:** Department of Obstetrics & Gynecology, Hallym University Dongtan Sacred Heart Hospital, Hwasungsi 18450, Kyeonggi-do, Korea; hycho@hallym.or.kr

**Keywords:** AMH, ovarian reserve, laparoscopy, hysterectomy, myomectomy

## Abstract

This prospective study aimed to investigate the impact of laparoscopic hysterectomy (LH) and laparoscopic myomectomy (LM) on ovarian reserve by comparing serum anti-Mullerian hormone (AMH) changes following surgery. Serum AMH levels were measured preoperatively (AMH0), and 7 days (AMH1), 2 months (AMH2), and 6 months (AMH3) after LH and LM in 79 premenopausal women (LH = 59; LM = 20). AMH0, AMH1, AMH2, and AMH3 were significantly higher in the LM group than in the LH group (*p* = 0.012, 0.001, 0.001, and 0.015, respectively). Since there are differences in indications between myomectomy and hysterectomy, logically, women who underwent myomectomy were younger and had higher AMH baseline levels. In addition, AMH changes at 7 days postoperatively from the baseline level were significantly decreased in the LH group compared to those in the LM group (*p* = 0.042). However, AMH changes at 2 months and 6 months postoperatively, compared to the baseline level, were not different between the two groups (*p* = 0.053 and 0.752, respectively). Moreover, the significant decrease in AMH (more than 60% decrease from the baseline level) was not different at 7 days, 2 months, and 6 months postoperatively between the two groups (*p* = 0.415, 487, and 0.364, respectively). Our data suggest that serum AMH levels were significantly decreased directly after LH, which suggests that LH may have adverse effects on ovarian reserve. However, mid-term follow-up showed that the damaged ovarian reserve in women who underwent LH may be partially restored in 6 months.

## 1. Introduction

Uterine fibroids (e.g., myomas and leiomyomas) are one of the most common types of benign monoclonal tumors of smooth muscle cells, occurring in approximately 50% of premenopausal women aged 35 years old [1]. Most women with asymptomatic uterine fibroids do not require further treatment. However, symptomatic uterine fibroids, including dysmenorrhea, menorrhagia, and compression of adjacent organs (e.g., urinary frequency and constipation), need to be treated. The main options for the surgical treatment of uterine fibroids are myomectomy or hysterectomy. Generally, hysterectomy has been recommended for women with multiple or large-sized fibroids and no desire for future fertility. However, a number of young women prefer myomectomy, not just because of the desire to preserve the uterus but because of the probable decreased ovarian function during the hysterectomy.

AMH is well known as a reliable and useful marker of ovarian reserve. The AMH level correlates with other markers of ovarian reserve, such as the antral follicle count or day 3 serum FSH concentration [2]. Moreover, the serum AMH level is not affected by the menstrual cycle, and it gradually declines with increasing age [3]. Not surprisingly, the AMH level is widely used as a predictor of ovarian reserve in clinical practice.

There is conflicting evidence regarding ovarian reserve after hysterectomy. Several studies supported that women who underwent ovary-sparing hysterectomy suffer from menopausal symptoms at a younger age because of the reduction in ovarian blood flow and follicular atresia [4,5]. A recent prospective study also supported that hysterectomy with bilateral salpingectomy significantly compromises ovarian reserve, and that the damage is more severe in younger women (<35 years old) [6].

In contrast, a large prospective study including 220 women who underwent abdominal hysterectomy with preservation of bilateral ovaries suggested that there were no significant changes in AMH, FSH, or estradiol at 6 and 12 months postoperatively [7]. Moreover, ovarian volume was increased and the pulsatility index of ovarian vessels was decreased at 12 months postoperatively, which indicates a possible increase in ovarian blood supply and preserved non-compromised ovarian function [7]. In comparison to abdominal hysterectomy, which ligates vessels with silk or Vicryl sutures, laparoscopic hysterectomy using electrothermal devices may have a higher risk of damaging ovarian reserve. A prospective study comparing laparoscopic hysterectomy vs. non-laparoscopic hysterectomy revealed that laparoscopic hysterectomy was an independent risk factor for a significant decrease in serum AMH (more than 30% from the baseline level) at 2 months postoperatively [8].

Nonetheless, there have been no reports comparing changes in ovarian reserve following laparoscopic hysterectomy to those of myomectomy. Therefore, we prospectively evaluated changes in ovarian reserve by the measurement of serum AMH, which represents ovarian reserve regardless of the menstrual period or patients’ physical condition, after laparoscopic hysterectomy (total laparoscopic hysterectomy (TLH) or laparoscopy-assisted vaginal hysterectomy (LAVH), study group), comparing those after laparoscopic myomectomy (LM, control group).

## 2. Materials and Methods

From 2019 to 2020, patients undergoing laparoscopic hysterectomy (TLH or LAVH) or LM for the treatment of uterine leiomyoma at Hallym University Medical Center, Korea, were recruited for this prospective study. Inclusion criteria were as follows: 1. women who were diagnosed with uterine leiomyoma by an imaging study (ultrasound, computed tomography, or magnetic resonance imaging); 2. premenopausal women (who have regular menstrual period (interval 25–90 days) within 2 years of study period).

The exclusion criteria were as follows: 1. postmenopausal women; 2. hormonal treatment within 3 months of surgery; 3. women who had any history of adnexal surgery and myomectomy; 4. women with any co-existence of endocrine disease; 5. women who had any exposure to irradiation or chemotherapeutic agents for the treatment of malignant disease.

This study was approved by the Hallym University Dongtan Sacred Heart Hospital Institutional Review Board (approval number 2016-01-004), and all recruited patients provided their informed written consent. Ovaries and fallopian tubes were all preserved during the hysterectomy (during the study period, salpingectomy was not routinely recommended for all women undergoing hysterectomy).

Indications of hysterectomy were as follows: 1. women with multiple leiomyomas (largest diameter more than 5 cm) or large-sized leiomyomas (largest diameter more than 10 cm) with no desire for fertility; 2. women with leiomyoma type 0~1 with no desire for fertility; 3. women who wanted a hysterectomy. Except for women who indicated wanting a hysterectomy, all women with leiomyomas underwent myomectomy. The smallest diameter of leiomyomas indicated for myomectomy was 5 cm.

All surgeries were performed by two experienced senior surgeons. During the laparoscopic procedure, from skin incision to trocar insertion, there were no differences between the two groups. Under general anesthesia, a laparoscopic pneumoperitoneum was induced by CO_2_ insufflations with an umbilical trocar. A rigid 0° 5 or 10 mm endoscope was used at the surgeon’s discretion. Under direct laparoscopic vision, two or three additional trocars (suprapubic and LLQ or RLQ area at the surgeon’s discretion) were inserted through lower abdominal incisions. Hysterectomies were performed in a conventional manner. In the TLH group, pelvic ligaments and utero-ovarian ligaments were dissected and cut using monopolar scissors and bipolar forceps. After the dissection of the bladder, colpotomy was performed using monopolar scissors. The vaginal cuff was closed by continuous suturing with 1-0 barbed suture (V-Loc™ 180 Absorbable Wound Closure, Covidein Healthcare, Mansfield, MA, USA). In the LAVH group, pelvic ligaments and utero-ovarian ligaments were dissected and cut using the monopolar scissors and bipolar forceps, which is a similar process to that of the TLH group. Then, colpotomy and extraction of the uterus were performed via the vagina. The vaginal cuff was closed by continuous suturing with 1-0 Vicryl. 

During the laparoscopic myomectomy, the capsule of the myoma was incised with monopolar scissors. Dissection and bleeding control of the myoma were performed using a bipolar forceps. The myometrium was repaired by continuous suturing with 1-0 barbed suture (V-Loc™ 180 Absorbable Wound Closure, Covidein Healthcare, Mansfield, MA, USA). Excised myomas were removed with in-bag manual morcellation.

### 2.1. Hormonal Assay

Serial measurements of serum AMH levels were undertaken in all patients: 1. AMH0 (serum AMH level within 2 weeks of surgery); 2. AMH1 (serum AMH level at day 7 postoperatively); 3. AMH2 (serum AMH level at 2 months postoperatively); 4. AMH3 (serum AMH level at 6 months postoperatively). Although AMH levels are well known to not be affected by hormonal periods, physical stress directly after surgery might affect AMH levels. Therefore, we set AMH1 as day 7 postoperatively to remove biases. Blood samples were obtained by venipuncture, and sera were extracted by a centrifuge. Serum AMH levels were measured by commercially available enzyme-linked immunosorbent assay kits (AMH Gen II ELISA) and reported as nanograms per milliliter with a detection limit of 0.006 ng/mL. All measurements were performed in the same reference laboratory.

In a prior study comparing laparoscopic hysterectomy vs. non-laparoscopic hysterectomy, a reduction of 30% in the serum AMH level from baseline was considered to be significant [8]. Over 40% of women had a more than 30% reduction in ovarian reserve after hysterectomy. Based on the prior study, we considered a reduction of 60% (2-fold of 30%) in serum AMH from the baseline level in each group.

### 2.2. Sample Size Calculation

This study aimed to show a 1.5 difference in the mean AMH between the two groups. Standard deviation was assumed to be similar in both groups. Based on two-tailed power calculation with 80% power, 0.05 alpha, and 1:1 allocation, 34 patients needed to be included in the trial to show this difference.

### 2.3. Statistical Analysis

Statistical analysis was performed using SPSS for Windows (version 21.0, SPSS Inc., Chicago, IL, USA). Dichotomous variables were compared by Fisher’s exact test or the chi-square test. Changes in hormone levels within the groups were analyzed by repeated measures analysis of variance (ANOVA). Continuous variables were compared by the t-test. The results are presented as mean ± SD. For all statistical tests, *p* < 0.05 was considered significant.

## 3. Results

A total of 79 women (LH = 59, LM = 20) agreed to participate in this study. The baseline data for all the patients are listed in Table 1.

The mean age of the LM group was younger than that of the LH group, which is associated with the general preference for the preservation of fertility in the younger age group (*p* = 0.014). Other factors, including parity and body mass index (BMI), were not different between the two groups. The mean largest diameter of myomas and mean number of myomas were not different between the LM and LH groups. According to the FIGO classification of the largest myoma location, there was no significant difference between the two groups.

Serial changes in the serum AMH of women following surgery are described in Table 2.

AMH0 in the LM group was significantly higher than that of the LH group (*p* = 0.012). Similarly, AMH1, AMH2, and AMH3 in the LM group were significantly higher than those of the LH group (*p* = 0.001, 0.001, and 0.015, respectively).

In addition, AMH changes at 7 days postoperatively from the baseline level were significantly decreased in the LH group compared to those in the LM group (*p* = 0.042). However, AMH changes at 2 months and 6 months postoperatively, compared to the baseline level, were not different between the two groups (*p* = 0.053 and 0.752, respectively). Percentage changes in AMH from the baseline level at 7 days, 2 months, and 6 months postoperatively were not different between the two groups (*p* = 0.576, 0.312, and 0.522, respectively). Moreover, the significant decrease in AMH (more than 60% decrease from the baseline level) was not different at 7 days, 2 months, and 6 months postoperatively between the two groups (*p* = 0.415, 487, and 0.364, respectively).

Serum AMH changes from the baseline level in the younger age group (younger than 45 years old) are described in Table 3. In the younger age group (LH = 28 and LM = 13), AMH0 and AMH3 were not different between the two groups (*p* = 0.076 and 0.131, respectively). However, AMH1 and AMH2 in the LM group were significantly higher than those of the LH group (*p* = 0.005 and 0.008, respectively). AMH changes at 7 days postoperatively from the baseline level were not different between the two groups (*p* = 0.195). Similarly, AMH changes at 2 months and 6 months postoperatively, compared to the baseline level, were not different between the two groups (*p* = 0.142 and 0.625, respectively). In addition, percentage changes in AMH from the baseline level at 7 days, 2 months, and 6 months postoperatively were not different between the two groups (*p* = 0.432, 0.429, and 0.621, respectively).

Serial changes in serum AMH levels following LM and LH are graphically depicted in Figure 1, Figure 2 and Figure 3. Serum AMH levels of women in the LH group showed a decrease at 7 days and 2 months postoperatively, but this was partially restored at 6 months postoperatively, while serum AMH levels of the LM group showed no significant changes throughout the study period.

Comparing serial AMH levels within each group by ANOVA revealed that AMH changes were not statistically significant in the LH (*p* = 0.468) and LM (*p* = 0.998) groups.

## 4. Discussion

There are concerns among gynecologic surgeons that ovary- and fallopian tube-sparing hysterectomies may accelerate menopausal symptoms. If this is true, women with uterine fibroids and high risk factors of primary ovarian insufficiency, such as a prior history of radiation or chemotherapy, genetic disorders (e.g., fragile X syndrome), autoimmune disease, and familial history of primary ovarian insufficiency, should opt for myomectomy rather than hysterectomy [9].

In general, women with uterine fibroids indicated for surgical removal prefer myomectomy rather than hysterectomy, since they believe that hysterectomy might negatively affect ovarian function. Theoretically, it is obvious that myomectomy may not have a negative effect on ovarian function. Nonetheless, there have been no reports comparing the AMH changes following myomectomy and hysterectomy. Therefore, we performed this study to provide clinical evidence which makes it easier to decide on the surgical procedure.

In our data, ovary- and fallopian tube-sparing hysterectomies have a negative impact on ovarian reserve at 7 days postoperatively compared to myomectomy. Similar to our data, a recent prospective longitudinal study evaluating serum AMH and FSH levels in 84 women supports the notion that a hysterectomy with bilateral salpingectomy significantly compromises ovarian reserve, and the damage is more severe in younger women [6]. The authors reported that postoperative changes in serum AMH were significantly higher in the younger age group (younger than 35 years old) (*p* < 0.0001). Moreover, postoperative serum AMH was significantly decreased (*p* < 0.0001) and FSH was elevated (*p* < 0.0001) from the baseline level [6]. However, they did not set the control group to compare the changes in AMH and FSH during the hysterectomy [6]. Another study comparing serum AMH levels after hysterectomy (*N* = 35; abdominal or laparoscopic-assisted vaginal hysterectomy) to those after myomectomy (*N* = 35; abdominal or laparoscopic myomectomy) revealed that the serum AMH level was significantly lower at 2 days and 3 months postoperatively following hysterectomy compared to the baseline level (*p* < 0.01) [10]. In the myomectomy group, the serum AMH level was significantly decreased at 2 days postoperatively but recovered to the baseline level at 3 months postoperatively [10]. The authors concluded that hysterectomy may have a more lasting adverse effect on ovarian reserve than myomectomy [10].

Many gynecologic surgeons are concerned about a decreasing ovarian arterial flow, which may have a negative impact on ovarian reserve following hysterectomy. A prospective study evaluating ovarian reserves after total abdominal hysterectomy with preservation of at least the unilateral ovary reported that the postoperative serum AMH level (at least 1 year after surgery) was decreased, and that the FSH level was increased, when compared to such levels of a healthy control group (*p* = 0.016 and 0.001, respectively) [11]. Hot flushing was also more frequently observed in women who underwent hysterectomy than in the control group (35% vs. 15.4%; *p* = 0.030) [11]. However, the resistive index and pulsatility index were not different between the two groups [11]. The authors concluded that total abdominal hysterectomy has a negative impact on ovarian remnant, which may be associated with the transfer of low-molecular weight substances between the uterine vein and ovarian artery rather than compromised ovarian blood flow [11]. Although this study showed an interesting viewpoint of ovarian reserve following hysterectomy with long-term follow-up (mean 2.5 years), the subject group was small, and there were no preoperative data of serum AMH and FSH levels [11].

In contrast to prior studies, a large-scale study including 220 women who underwent total abdominal hysterectomy (TAH) with the preservation of both ovaries reported that serum AMH, FSH, and estradiol levels at 12 months postoperatively were not significantly different to baseline levels [7]. In addition, ovarian volumes were increased and ovarian pulsatility indices were decreased at 12 months postoperatively, which suggests a possible increase in ovarian blood supply and preserved non-compromised ovarian function after surgery [7,12]. Similarly, several studies have reported that there were no significant changes in ovarian hormones and ovarian artery blood flows following ovarian-preserving hysterectomy [12,13]. A prospective study including 22 women who underwent total abdominal hysterectomy reported that serum AMH levels at 4 months postoperatively were decreased by 28.5% compared to the baseline level, but this was not statistically significant (*p* = 0.26) [14].

Among the discussions regarding whether ovarian-preserving hysterectomy promotes ovarian damage, and what the mechanism is, there have been very few studies evaluating ovarian reserve after laparoscopic hysterectomy. Most studies evaluated the effect of abdominal or vaginal hysterectomy on ovarian reserve [7,11,12]. A prospective study comparing LH vs. non-laparoscopic hysterectomy revealed that LH was an independent risk factor for a significant decrease in serum AMH at 2 months postoperatively (hazard ratio 4.147, 95% confidence interval 1.139–15.097) [8]. The main difference of laparoscopic hysterectomy from non-laparoscopic hysterectomy is the use of electrothermal vessel ligation. During vaginal or abdominal hysterectomy, vessel sealing is usually performed with suture ties using Vicryl or silk. During LH, electrothermal vessel ligation is performed in laparoscopic hysterectomy, which can directly damage adjacent ovarian tissue and vessels, thereby accelerating follicular depletion and earlier menopause.

There have been several reports evaluating ovarian reserves following laparoscopic ovarian cystectomy [15,16,17,18]. Most of those studies reported that ovarian reserves decreased abruptly after surgery, although they recovered completely after one year of surgery [15,16]. The main reason for decreasing ovarian reserves is thought to be non-selective bleeding control using a bipolar device [17]. A recent cohort study comparing changes in serum AMH following total laparoscopic hysterectomy (TLH, *N* = 40) and laparoscopic supracervical hysterectomy (LSH, *N* = 43) showed a significant decrease in serum AMH levels at 1 and 4 months postoperatively in both groups (*p* < 0.001) [19]. Similarly, our data show an abrupt decrease in ovarian reserves in the LH group until 2 months postoperatively, which was partially recovered at 6 months postoperatively.

Our study is the first prospective study evaluating ovarian reserves after LH in comparison to LM. According to our data, the mean age in the LM group was approximately 2.8 years younger than that of the LH group, which may suggest the general preference for the preservation of fertility in younger age groups. In our study, AMH0 in the LM group was significantly higher than that of the LH group. Since there are differences in indications between myomectomy and hysterectomy, logically, women who underwent myomectomy were younger and had higher AMH baseline levels. Despite the younger age of participants in the LM group, changes in AMH levels at 2 months and 6 months postoperatively were not significantly different between the two groups. In addition, the rate of significantly decreased ovarian reserve (more than 60% decreases from the baseline level) was similar between the two groups. These findings were not different in the sub-analysis of the younger age group (younger than 45 years old). Therefore, the differences in serum AMH between the two groups during the study period will be further decreased after age adjustment.

There are some limitations in our study.

We did not consider the location and size of all uterine myomas in evaluating ovarian reserve of the LM group. It can be considered that myomas near the ovary (e.g., broad ligament) might have a negative impact on ovarian reserve during LM. In our study, we simply investigated the association of serum AMH changes and the FIGO type of the largest myoma in the LM group. There was no significant differences in serum AMH between the two groups. However, this is not considered to be an important issue as this is a study to find out how much the ovarian reserve function changes after LH compared to LM in patients with myomas.We did not compare blood loss during the surgery. Heavy bleeding during the surgery can be a risk factor in the reduction in ovarian reserve. In future study, blood loss might be included in the compared variables.We included a small sample of study subjects, specifically in the LM group. Based on our data, more large-scaled study will be required.

In conclusion, LH may have a negative impact on ovarian reserve directly after surgery, but the damaged ovarian reserve can be partially restored with mid-term follow-up. Our data will provide valuable information to the gynecologist and premenopausal women when they make a decision on the surgical treatment for uterine myomas. Women with high risk factors of primary ovarian failure should be cautiously considered in decision making.

## Figures and Tables

**Figure 1 jcm-10-03077-f001:**
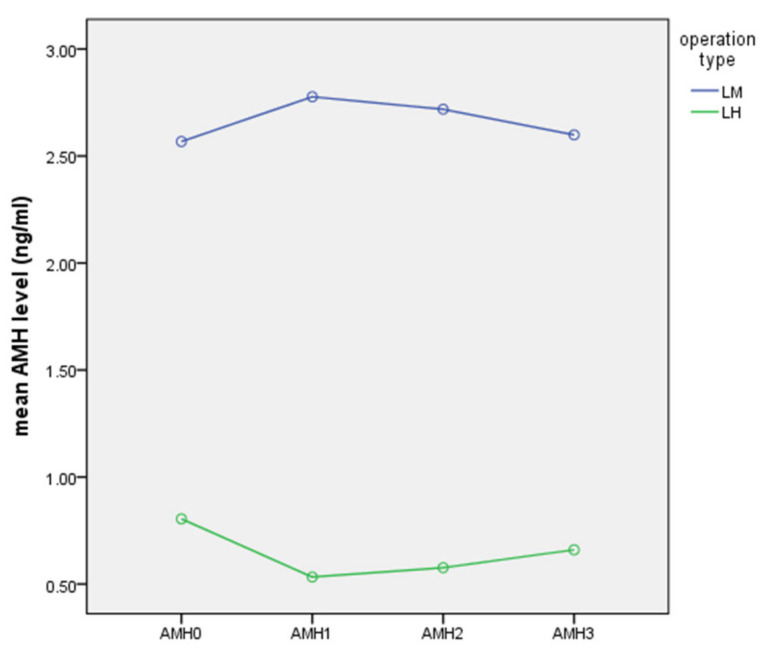
Serial changes in serum AMH following LM or LH (ng/mL).

**Figure 2 jcm-10-03077-f002:**
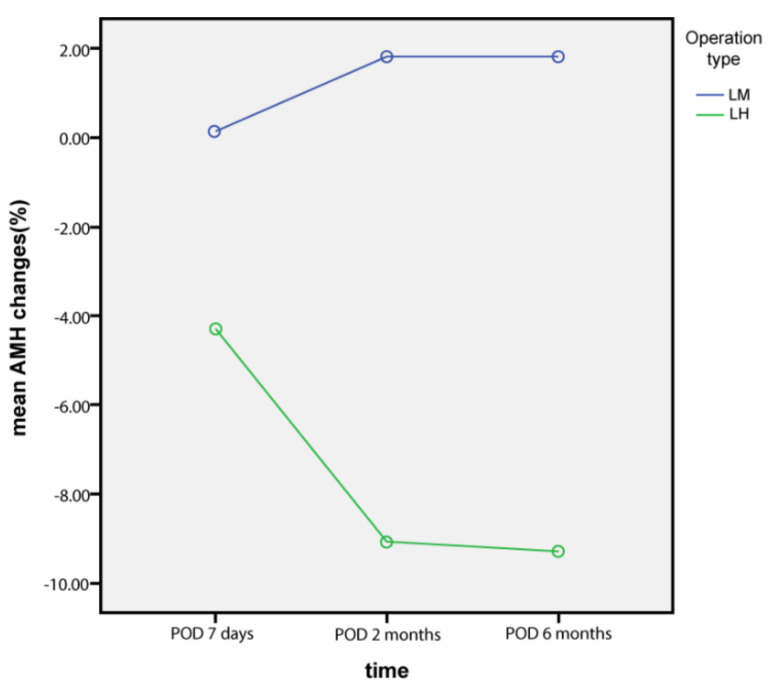
AMH changes from baseline levels (%).

**Figure 3 jcm-10-03077-f003:**
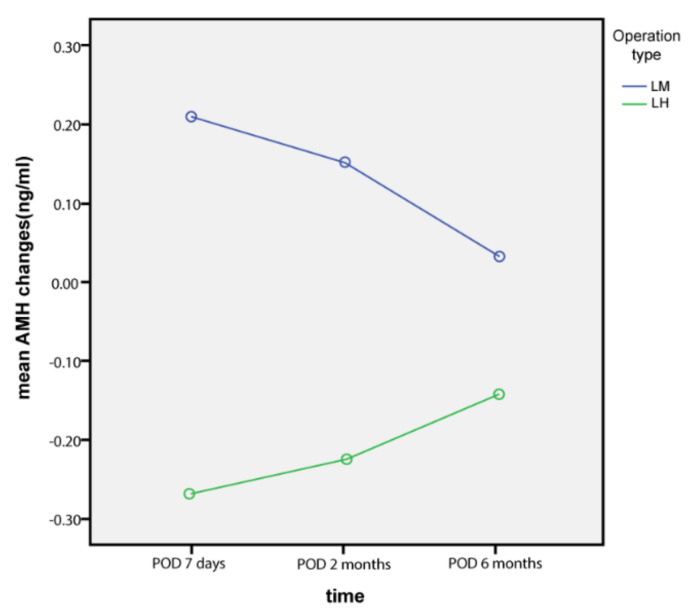
AMH changes from baseline levels (ng/mL).

**Table 1 jcm-10-03077-t001:** Baseline data.

	LM (*N* = 20)	LH (*N* = 59)	*p*-Value
Age (years)	41.3 ± 5.76	44.1 ± 3.73	0.014 *
Parity	1.3 ± 1.03	1.6 ± 0.77	0.157
BMI	24.8 ± 4.41	24.4 ± 5.32	0.682
Largest diameter of myoma	8.4 ± 2.81	7.6 ± 3.04	0.142
Number of myomas	2.0 ± 1.70	2.6 ± 1.91	0.151
FIGO classification of largest myoma			0.254
Type 0–2	3	10	
Type 3–5	13	40	
Type 6–7	4	9	

LM, laparoscopic myomectomy; LH, laparoscopic hysterectomy; BMI, body mass index; FIGO, International Federation of Gynecology and Obstetrics. * *p*-value < 0.05

**Table 2 jcm-10-03077-t002:** AMH changes compared to the baseline levels.

	LM (*N* = 20)	LH (*N* = 59)	*p*-Value
AMH levels			
Baseline level (AMH0)	2.4 ± 2.72	1.1 ± 1.72	0.012 *
7 days postoperatively (AMH1)	2.6 ± 3.28	0.8 ± 1.25	0.001 *
2 months postoperatively (AMH2)	2.6 ± 3.11	0.8 ± 1.39	0.001 *
6 months postoperatively (AMH3)	2.6 ± 3.60	0.7 ± 1.34	0.015 *
AMH changes compared to baseline level			
7 days postoperatively (AMH1)	0.2 ± 1.44	−0.3 ± 0.64	0.042 *
2 months postoperatively (AMH2)	0.1 ± 1.31	−0.3 ± 0.62	0.053
6 months postoperatively (AMH3)	0.0 ± 1.82	−0.1 ± 0.48	0.752
AMH changes compared to baseline level (%)			
7 days postoperatively (AMH1)	−2.6 ± 46.16	−10.5 ± 53.46	0.576
2 months postoperatively (AMH2)	−1.1 ± 51.61	−15.1 ± 49.26	0.312
6 months postoperatively (AMH3)	1.8 ± 50.81	−9.3 ± 64.09	0.522
AMH1 decrease ≥ 60% (11/79)	2 (10.0)	9 (15.3)	0.415
AMH2 decrease ≥ 60% (15/79)	3 (15.0)	12 (20.3)	0.487
AMH3 decrease ≥ 60% (7/44)	2 (10.5)	5 (20.0)	0.364

LM, laparoscopic myomectomy; LH, laparoscopic hysterectomy; AMH, anti-Mullerian hormone; *, *p*-value < 0.05.

**Table 3 jcm-10-03077-t003:** AMH changes compared to the baseline levels in the younger age group (younger than 45 years old).

	LM (*N* = 13)	LH (*N* = 28)	*p*-Value
AMH levels			
Baseline level (AMH0)	3.5 ± 1.56	1.8 ± 2.22	0.076
7 days postoperatively (AMH1)	3.8 ± 3.60	1.3 ± 1.57	0.005 *
2 months postoperatively (AMH2)	3.7 ± 3.39	1.4 ± 1.82	0.008 *
6 months postoperatively (AMH3)	3.6 ± 3.97	1.6 ± 1.95	0.131
AMH changes compared to baseline level			
7 days postoperatively (AMH1)	0.3 ± 1.78	−0.4 ± 0.85	0.195
2 months postoperatively (AMH2)	0.2 ± 1.60	−0.4 ± 0.77	0.142
6 months postoperatively (AMH3)	0.2 ± 2.21	−0.2 ± 0.80	0.625
AMH changes compared to baseline level (%)			
7 days postoperatively (AMH1)	3.9 ± 43.89	−8.3 ± 46.36	0.432
2 months postoperatively (AMH2)	2.85 ± 47.08	−8.3 ± 48.56	0.429
6 months postoperatively (AMH3)	−3.1 ± 52.86	11.9 ± 91.16	0.621
AMH1 decrease ≥ 60% (2/41)	1 (7.7)	1 (3.6)	0.598
AMH2 decrease ≥ 60% (4/41)	1 (7.7)	3 (10.7)	0.598
AMH3 decrease ≥ 60% (3/22)	2 (15.4)	1 (11.1)	0.611

LM, laparoscopic myomectomy; LH, laparoscopic hysterectomy; AMH, anti-Mullerian hormone; *, *p*-value < 0.05.

## Data Availability

The datasets generated during and/or analyzed during the current study are available from the corresponding author on reasonable request.

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
