# Peer review of "Comparison of Postoperative Ovarian Reserve Function Following Laparoscopic Hysterectomy and Laparoscopic Myomectomy: A Prospective Comparative Pilot Study"

_jcm, 2021, doi:10.3390/jcm10143077_

Round 1
Reviewer 1 Report
None
Author Response
Thank you for your good response.
Reviewer 2 Report
The authors have revised correctly according to the reviewer's suggestions.
Author Response
Thank you for your good response.
Reviewer 3 Report
I think this is an interesting study, especially that the laparoscopic hysterectomy was performed in one of the study group. In most previous publications the authors had examined the ovarian reserve after vaginal or abdominal hysterectomy. You should write more about the difference between abdominal and laparoscopic hysterectomy on ovarian reserve: the role of electrothermal devices or Vicryl sutures during TAH on ovarian reserve (the data in the literature) in the discussion section. The difference in age of the two groups and the small number of patients taking part in the study are the weakest point of this paper. Taking into account the small number of patients who took part (n = 79) in the study, I propose to add in the title- a prospective comparative pilot study.In the Material and Method section, I propose to add the size of myoma as a surgery indication - the smallest size of the myomas in patients who underwent myomectomy
Reviewer 4 Report
Summary and broad comments
This is an interesting prospective comparative monocentric study in which changes in ovarian reserve following laparoscopic hysterectomy (LH group) were compared to those of laparoscopic myomectomy (LM group), by comparing serum AMH serial changes. The aim of this study is to provide evidence to help in choosing the best surgical strategy in premenopausal women with uterine fibroid.
The question is original, since there are currently no reports in which changes in ovarian reserve were compared in these two groups, and the article is written in an appropriate way but there are minor inaccuracies in the presentation of the data (e.g., P value in Table 2 compared to P value in the abstract and results, see specific comments). Data and conclusions are interesting and the subanalysis among the younger patients of the two groups is a strength; nevertheless there are some limitations: small sample size (only 79 premenopausal women: 59 LH , 20 LM); in the LM group the distance between myomas and ovaries is unknown; the statistical significance of the serial changes in serum AMH levels of the same group was not evaluated.
Specific comments
Abstract
The abstract is clear and summarizes the main results of the study, but there are some inaccuracies, supposedly misprints: in line 17, 21,22,23 and 25 the reported P value is different from those present in Table 2.
Introduction
The introduction provides sufficient background and includes interesting references, however it might be useful to explain why the authors chose AMH for assessing ovarian reserve and to stress on the health implications of an earlier menopause.
Materials and methods
About eligibility criteria: is age an inclusion criteria or not? Was the premenopausal status assessed in any way? Furthermore, the authors do not specify if some patient had premature ovarian failure familiarity or history of myomectomy. It might be useful to clarify these aspects.
In line 93 and 96 the term “gonadal vessels” could be wrongly understood as infundibulopelvic ligament, probably it would be clearer to use the term utero-ovarian ligament.
Hormonal assay
In line 108 AMH3 was defined as serum AMH level at postoperative 3 months but in the abstract and results the time point was set to 6 months, supposedly a misprint.
Statistical analysis
It would be useful to specify why a reduction of 60% in serum AMH level is considered significant and what is the lowest number of subjects to be recruited to achieve the statistical significance.
Results
In line 142 the reported P value (AMH0 and AMH3) is different from those present in Table 2.
Figures 1-3, in which the serial changes in AMH level of each group are graphically depicted, are interesting, however it would be supportive to evaluate the statistical significance of the serial changes in the same group and not only between the two groups.
Discussion
The discussion is clear and summarizes the literature well. However, the analysis of the results, their impact and the limitations of the present study could be improved.
